# Epigenetic Methylation Changes in Pregnant Women: Bisphenol Exposure and Atopic Dermatitis

**DOI:** 10.3390/ijms25031579

**Published:** 2024-01-27

**Authors:** Seung Hwan Kim, So Yeon Yu, Jeong Hyeop Choo, Jihyun Kim, Kangmo Ahn, Seung Yong Hwang

**Affiliations:** 1Department of Bio-Nanotechnology, Hanyang University, Ansan 15588, Republic of Korea; kandoli1@daum.net; 2Department of Molecular & Life Science, Hanyang University, Ansan 15588, Republic of Korea; yusso3027@naver.com (S.Y.Y.); cnwjdguq@naver.com (J.H.C.); 3Department of Pediatrics, Samsung Medical Center, Sungkyunkwan University School of Medicine, Seoul 06351, Republic of Koreakmaped@gmail.com (K.A.); 4Department of Health Sciences and Technology, Samsung Advanced Institute for Health Sciences & Technology, Seoul 06355, Republic of Korea; 5Department of Medicinal and Life Sciences, Hanyang University, Ansan 15588, Republic of Korea; 6Department of Applied Artificial Intelligence, Hanyang University, Ansan 15588, Republic of Korea

**Keywords:** bisphenol A, bisphenol S, bisphenol F, atopic dermatitis, DNA methylation, JAK-STAT signaling pathway, PI3K-AKT signaling pathway

## Abstract

Bisphenol is a chemical substance widely used in plastic products and food containers. In this study, we observed a relationship between DNA methylation and atopic dermatitis (AD) in the peripheral blood mononuclear cells (PBMCs) of pregnant women exposed to bisphenol A (BPA) and its alternatives, bisphenol S (BPS) and bisphenol F (BPF). DNA methylation is an epigenetic mechanism that regulates gene expression, which can be altered by environmental factors, and affects the onset and progression of diseases. We found that genes belonging to the JAK-STAT and PI3K-AKT signaling pathways were hypomethylated in the blood of pregnant women exposed to bisphenols. These genes play important roles in skin barrier function and immune responses, and may influence AD. Therefore, we suggest that not only BPA, but also BPS and BPF, which are used as alternatives, can have a negative impact on AD through epigenetic mechanisms.

## 1. Introduction

With advances in industrial technology, many convenient chemical substances have been developed for daily use. Moreover, as these chemical substances have become common in our daily lives, their risks have increased. Therefore, it is important to identify and analyze the environmental factors that affect human health and disease. The concept proposed for this purpose is the exposome, and its importance has been emphasized [1,2].

The exposome is defined as the totality of various external factors to which an individual is exposed throughout their lifetime, and includes biological responses related to various factors such as environmental and lifestyle factor [1,3,4,5]. The exposome is a concept that contrasts with the genome [6], and consists of three domains [7]: the general external domain, such as social status, the surrounding environment, and climate; the specific external domain, such as chemical pollutants and pathogens; and the biological internal domain, such as metabolism, oxidative stress, and aging. Methods for studying exposomes include cohort studies, environment-wide association studies, and omics [6,8,9].

Bisphenol A (BPA), bisphenol F (BPF), and bisphenol S (BPS) are chemical substances used to prepare polycarbonate and epoxy resins. Polycarbonate is transparent and hard and is used as a consumer product for water bottles, sports equipment, etc. BPA is a widely used organic synthetic compound with two hydroxyl phenol groups and the chemical formula (CH_3_)_2_C(C_6_H_4_OH)_2_. However, BPA can enter the body through various routes, such as inhalation and skin contact, and shows various toxic effects, including AD [10,11]. Therefore, BPF and BPS can be used as alternatives to BPA [12]. BPF is an organic synthetic compound with two hydroxyl phenol groups and the chemical formula (CH_3_)_2_C(C_6_H_3_ (OH)_2_)_2_. BPS is an organic synthetic compound with two sulfone phenol groups and the chemical formula (CH_3_)_2_C(C_6_H_4_SO_2_)_2_. However, research on the human exposure levels and health effects of BPF and BPS remains insufficient.

Environmental pollutants such as bisphenols are harmful chemicals that affect human health and can enter the body through air, soil, and food contamination. Exposure to these environmental pollutants can cause abnormalities in the immune system and inflammation in organs such as the skin, respiratory system, and digestive system, which can lead to environmental diseases. Environmental diseases include asthma, atopy, allergy, chronic obstructive pulmonary disease (COPD), chronic sinusitis, etc.; these are diseases that many people suffer from worldwide. AD is a common chronic inflammatory skin disease affecting more than 20% of children and 10% of adults worldwide [13]. AD occurs due to defects in skin barrier function, immune regulation disorders, and complex interactions between environmental and infectious factors [14]. AD is characterized by recurrent itching and localized eczema, and many patients have allergies, such as allergic asthma, allergic rhinitis, and food allergies [15]. AD is a heterogeneous disease that shows varied clinical phenotypes, and can be classified into extrinsic and intrinsic subtypes depending on the morphology and distribution of lesions, serum IgE levels, types of allergic reaction, etc. [16,17]. Distinguishing AD subtypes can help understand and improve the etiology, diagnosis, prognosis, treatment, and management of AD. However, AD is generally known as a Th2-based disease, and the distinction between subtypes is not permanent as intrinsic AD can become extrinsic AD [18].

DNA methylation is a phenomenon in which a methyl group is attached to the cytosine base of a DNA molecule. It mainly occurs in the region where cytosine and guanine are connected, and is called a CpG dinucleotide. DNA methylation is involved in important biological processes such as gene expression regulation, chromosome stability, X-chromosome inactivation, and gene imprinting, and affects various life phenomena such as cell differentiation, development, and aging [19]. Unlike genetic mutations, DNA methylation does not change the DNA sequence itself, but can be reversibly altered by environmental factors inside and outside the cell [20]. The reversibility of DNA methylation results in cell adaptability [21]. DNA methylation has a specific pattern that only appears in certain tissues or cells, and can be used to identify the identity and state of cells [22]. DNA methylation is associated with various diseases. Especially in patients with AD, the methylation of skin-barrier-related genes such as filaggrin and immune response-related genes such as cytokines decreases or increases, causing damage and inflammation of the skin barrier [23,24]. Therefore, DNA methylation is receiving considerable attention as a biomarker for diagnosis, prognosis, treatment response, and other diseases.

In this study, we aimed to observe the correlation between BPS and BPF, which are used as alternatives to BPA, and AD from the perspective of exposome. To do this, we used the GREEN (Growing children’s health and Evaluation of Environment) cohort, which was recruited for the study of environmental diseases caused by environmental pollutants, to assess the exposure levels of bisphenols in adult women and to analyze the correlation between methylation changes and atopic diseases.

## 2. Results

### 2.1. Information of Participants for the Analysis of Differentially Methylated Regions (DMRs)

We divided the participants into atopic and non-atopic groups and compared their information by group (Table 1). Of 117 participants, 19 had atopy. Most participants were in their 30s, with an average age of approximately 33.9 years, and most were non-smokers. The average exposure levels of BPA, BPF, and BPS for all participants were 1.56 μg/g cr., 0.78 μg/g cr., and 0.30 μg/g cr., respectively, which were slightly higher than the average exposure levels of BPA, BPF, and BPS for adults according to the results of the National Environmental Health Survey in Korea, which were 1.08 μg/g cr., 0.19 μg/g cr., and 0.19 μg/g cr.

We separated the case and control groups for the DMR analysis and checked the exposure levels of each substance by group (Table 2). The BPA group consisted of three atopic patients with high BPA exposure and an average age of 38.7 years. The control group consisted of 72 non-atopic participants with low exposure and an average age of 33.9 years. The mean BPA exposure level was 2.5 μg/g cr. in the case group and 0.6 μg/g cr. in the control group, which was lower than that in the case group. In the BPF exposure group, the case group consisted of six atopic patients with an average age of 32.5 years and an average exposure level of 1.1 μg/g cr., whereas the control group consisted of 75 non-atopic participants with an average age of 33.9 years and an average exposure level of 0.1 μg/g cr. Finally, in the BPS exposure group, the case group consisted of five atopic patients with an average age of 33.8 years and an average exposure level of 1.2 u/g cr., whereas the control group consisted of 74 non-atopic participants with an average age of 33.5 years and an average exposure level of 0.1 μg/g cr.

### 2.2. Testing the Independence of Variables

Before conducting the analysis, the independence of the five variables, age, smoking status, BPA exposure level, BPF exposure level, and BPS exposure level, was examined by calculating the Spearman correlation between each variable (Table 3). Spearman correlations range from −1 to 1: the closer the value is to 0, the less correlation there is between the two variables; the closer the value is to 1, the stronger the positive correlation; and the closer the value is to −1, the stronger the negative correlation [25]. All correlations were below 0.3, indicating that the correlation between the two variables was weak or almost nonexistent.

The *p*-value was calculated to test the significance of the correlation coefficient (Table 4). The *p*-value for all variables except age and smoking status was greater than 0.05, indicating that each variable did not affect the other variables.

### 2.3. DMR Analysis According to Bisphenol Exposure

DMR analysis was performed by comparing the case and control groups grouped by each substance and setting the window size to 100 bp. We analyzed with a cutoff of |FC| > 2, *p*-value > 0.05, and summarized the DMR analysis results in a table (Appendix A). The BPA exposure group included 80,072 DMRs, of which 81 were hypermethylated and 79,991 were hypomethylated. The BPF and BPS exposure groups had 92,573 and 61,367 DMRs, respectively, of which 160 and 3056 were hypermethylated and 92,413 and 58,311 were hypomethylated, respectively (Table 5).

### 2.4. Bisphenol and AD

We matched the regions summarized using DMR analysis with the genes, summarized the methylation levels of the genes, and then compared them with the AD-related genes organized by the database (Table 6). There were 390, 356, and 303 AD-related genes in the BPA, BPF, and BPS exposure groups, respectively. There were 1, 2, and 14 hypermethylated genes in the BPA, BPF, and BPS exposure groups, respectively. There were 389, 354, and 289 hypomethylated genes in the BPA, BPF, and BPS exposure groups, respectively.

The numbers of genes derived from DMR analysis are shown in a Venn diagram (Figure 1). Among the hypermethylated genes, no common genes were expressed after BPA, BPF, and BPS exposure (Figure 1a). Most of the hypomethylated genes were included in all BPA, BPF, and BPS exposure groups. Among them, IL4 (BPA logFC −1.76, BPF logFC −1.75, BPS logFC −1.62), one of the cytokines known to be associated with atopy [26], was included, and IL13RA1 (BPA logFC −2.67, BPF logFC −1.60, BPS logFC −2.08), which is known to be expressed in keratinocytes [27], was also commonly hypomethylated in all three exposure groups (Figure 1b).

We verified the association between 201 AD-related genes derived from the three bisphenol exposure results and atopy. Among the common pathways, we confirmed that the JAK-STAT and PI3K-AKT signaling pathways are both influenced by IL4 and are related to AD [28,29,30,31,32,33]. From the DMR analysis results, we identified various genes that were commonly hypomethylated in the BPA, BPF, and BPS exposure groups, including IL4, JAK1 (BPA logFC −2.77, BPF logFC −2.13, BPS logFC −1.90), IL13RA1, and STAT3 (BPA logFC −2.41, BPF logFC −1.91, BPS logFC −1.50), which belong to the PI3K-AKT and JAK-STAT signaling pathways (marked in red in Figure 2). BCL2 (BPA logFC −2.59, BPF logFC −1.09, and BPS logFC −1.65), which is involved in the PI3K-AKT signaling pathway, was also found to be common in the three bisphenol exposure groups and was overexpressed in patients with AD [34].

In addition, various genes involved in the PI3K-AKT and JAK-STAT signaling pathways showed changes in methylation (Table 7). IL33, another cytokine gene associated with AD [35] besides IL4, was hypomethylated in both BPA and BPF exposure groups, as well as in TYK2, which phosphorylates and activates STAT3 in the JAK-STAT pathway. JAK3, which is phosphorylated along with JAK1 in type 1 IL4-R, was also hypomethylated in both BPA and BPS exposure groups.

## 3. Discussion

AD is an inflammatory skin disease with increasing prevalence worldwide [36]. The cause of AD is a complex interaction between genetic and environmental factors. One of the suggested environmental factors is BPA exposure. BPA is a chemical substance widely used in plastic products and food containers [37], which can affect the immune system and skin barrier function in the body [38]. Several studies have shown that populations exposed to BPA have a higher risk of atopy. As concerns regarding the safety of BPA increased, BPF and BPS were used as alternatives. However, research on BPF and BPS is relatively scarce compared with that on BPA, especially from an epigenetic perspective. Epigenetics studies the various mechanisms that regulate gene expression without changing the gene sequence, which can be altered by environmental factors that affect the onset and progression of diseases.

This study observed changes in DNA methylation in the PBMC of pregnant women exposed to BPA, BPF, and BPS. Of the 117 pregnant women, 19 had AD. The body exposure level of bisphenols was the highest for BPA, and the average exposure levels of all participants were slightly higher than the results of the National Environmental Health Survey. This difference may be due to the fact that the National Environmental Health Survey surveyed all adults, while this study only targeted women in their 30s who were of reproductive age.

For the DMR analysis, we separated the case group of AD patients with high exposure and the control group of non-AD patients with low exposure. We confirmed that IL4, a cytokine known to be associated with atopy, IL13RA1, known to be expressed in keratinocytes, and BCL2, known to be overexpressed in patients with AD, were commonly hypomethylated in all BPA, BPF, and BPS exposure groups. In addition, we confirmed that IL33, TYK2, JAK1, and JAK3 are hypomethylated. This suggests that these genes were relatively overexpressed, and we confirmed that these genes belong to the JAK-STAT and PI3-AKT signaling pathways.

There are two types of IL-4 receptors in the JAK-STAT and PI3K-Akt signaling pathways. When IL-4Rα binds to γc, it becomes IL-4R type 1, and when it binds to IL-13Rα1, it becomes IL-4R type 2 [39]. Type 1 IL-4R binds to IL-4 and phosphorylates insulin receptor substances (IRSs) through JAK1 phosphorylation [40]. This induces the activation of AKT signalling, including that of PI3K, and affects the growth, survival, and proliferation of keratinocytes [29]. Type 2 IL-4R binds to IL-4 and phosphorylates JAK1 and TYK2, which, in turn, phosphorylate and dimerize STAT6 and STAT3, respectively. STAT6 and STAT3 move to the nucleus and affect skin barrier dysfunction, which is known to be involved in the production and regulation of Th2 cytokines, major factors in the development of AD. Bisphenols can worsen the symptoms of AD by reducing the methylation of these genes and increasing their expression.

This study shows that both BPA and its substitutes, BPF and BPS, can affect AD through epigenetic mechanisms. However, this study had some limitations. First, the cohort used in this study only targeted pregnant women; therefore, the body exposure level to bisphenols may differ from that of adults in general. Second, this study only used DMR analysis; therefore, other factors involved in gene expression regulation were not considered. Recent studies have used multi-omics instead of single-omics [41,42]. Therefore, future studies should elucidate the complex gene expression network by integrating DMR analysis with other omics data [43]. This integrated analysis reflects the actual situation of human exposure to bisphenols, identifies the biochemical changes that occur in the body, and reveals the causal relationship and risk factors of disease onset.

## 4. Materials and Methods

### 4.1. Human-Derived Samples

This study conducted experiments on pregnant women from the GREEN (Growing Children’s Health and Evaluation of Environment) cohort (IRB #2016-12-111) recruited between 2017 and 2021. The GREEN cohort consisted of 151 pregnant women and 99 infants, and blood and urine samples were collected from pregnant women. PBMCs were collected from the blood, and gDNA was obtained to perform MeDIP-seq; urine samples were analyzed for urinary concentrations of BPA, BPF, and BPS using liquid chromatography mass spectrometry (LC/MS/MS) by Smartive Co., Ltd. (Hanam City, Gyeonggi-do, Republic of Korea), and then corrected for creatinine.

### 4.2. Testing the Independence of Variables

In this study, the Spearman correlation was calculated to test the independence of the variables used. The Spearman correlation is a statistical indicator that shows the strength and direction of the relationship between the ranks of two variables, ranging from −1 to 1. The *p*-value was calculated to test the significance of the correlation coefficient. The correlation analysis was performed using the SPSS (Ver. 27) program.

### 4.3. Classification of Exposure Groups for BPA, BPF, and BPS

In this study, we used the exposure levels of BPA, BPF, and BPS measured in the cohort. The exposure levels of BPA, BPF, and BPS may be related to the occurrence of AD; therefore, we classified them into control and case groups according to the exposure levels of each substance. The upper 25% of the exposure level was classified as a high-exposure group and the lower group as a low-exposure group (BPA: 1.685 μg/g creatinine, BPF: 0.445 μg/g creatinine, and BPS: 0.178 μg/g creatinine). For each substance, individuals with AD and belonging to the high-exposure group were designated as case groups and individuals without AD and belonging to the low-exposure group were designated as control groups. In this classification, the low-exposure group also included values that the device could not measure; >Limit of Detection (>LOD) was calculated using Equation (1) according to the reference of Korea Occupational Safety and Health Agency (KOSHA). The LOD values of BPA, BPF, and BPS were 0.052, 0.008, and 0.025, respectively.
(1)$$[>LOD]=\frac{LOD}{\sqrt{2}}$$

According to the equation, the >LOD values of BPA, BPF, and BPS were replaced with 0.03677 μg/g cr., 0.056569 μg/g cr., and 0.017678 μg/g cr., respectively.

### 4.4. MeDIP-Seq (DNA Preparation, Library Construction, and Sequencing)

Blood was collected using heparin tubes, and PBMCs were separated from the collected blood. gDNA was extracted from the PBMCs using an Invitrogen PureLink Genomic DNA Mini Kit (ThermoFisher, Waltham, MA, USA). The purity of the extracted gDNA was measured using a Nanodrop (ThermoFisher) and quantified using a Qubit (ThermoFisher). DNA fragmentation was performed using a Bioruptor (Diagenode, Seraing, Belgium) and immunoprecipitation was performed using an antibody. An adapter-attached library was constructed using the adaptor index included in the TruSeq RNA UD Indices (Illumina, San Diego, CA, USA). PCR was performed to amplify DNA with an adaptor index attached, and only 300–350 bp size were selected using Pippin Prep (Sage Science, Beverly, MA, USA). The constructed library was sequenced with 2 × 75 bp read length of paired-end using Nextseq500 (Illumina).

### 4.5. Differentially Methylated Regions (DMR) Analysis

We performed a quality check on the raw data using FastQC (Ver. 1.11.9) and trimmed the raw data using Trimmomatic (Ver. 0.40) program. We used TruSeq3-PE.fa, which was provided by Trimmomatic for the adapter sequence during the trimming process. We created an index for the hg38 reference genome using bowtie2 (Ver. 2.4.2) and mapped them. We converted the SAM format derived later into the BAM format using SAMtools (Ver. 1.11.0), and sorted the files based on mapped position information. Duplicate sequences were removed from the sorted files using Picard (Ver. 2.25.0). DMR analysis was performed using MEDIPS (ver. 1.42.0), a R (Ver. 4.3.1) package (*p*-value < 0.05, |FC| > 2). We analyzed only autosomes and the X chromosome because we performed the analysis for adult women, divided the window size into 100 bp, and proceeded and integrated the continuous position. We set the DMR annotation as ‘GENE’ and marked it as a gene containing the position.

### 4.6. Functional Analysis

We obtained a list of 842 genes related to atopy from the Comparative Toxicogenomics Database (CTD) and DisGeNET DB. We also compared the genes derived from DMR analysis with AD-related genes and analyzed the functions and cellular pathways of the derived genes using the KEGG PATHWAY Database of the Database for Annotation, Visualization and Integrated Discovery (DAVID) tool.

### 4.7. Ethical Considerations

All patients provided informed written consent for participation in the study, as regulated by local law. This study was approved by the Institutional Review Board (IRB #2016-12-111) of Samsung Seoul Hospital, and written consent was obtained from the parents who participated in this study.

## 5. Conclusions

In this study, we performed DMR analysis with pregnant women to investigate the correlation between bisphenols (BPA, BPF, and BPS) and AD. The results showed the hypomethylation of various genes known to affect AD in BPA, BPF, and BPS, and confirmed that these genes were involved in the JAK-STAT and PI3K-AKT signaling pathways. This study confirmed that BPA, BPF, and BPS can negatively affect atopy.

## Figures and Tables

**Figure 1 ijms-25-01579-f001:**
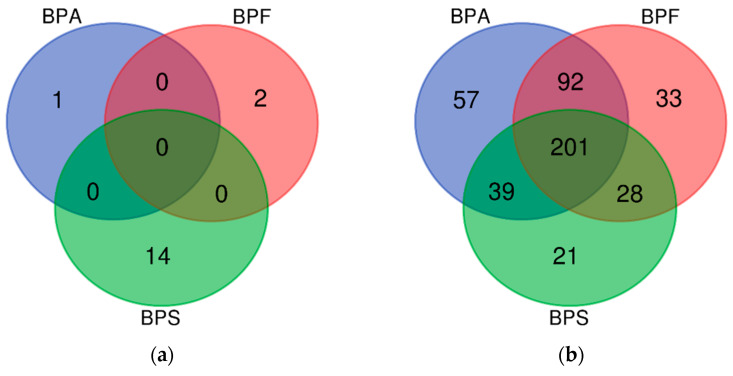
Venn diagram of genes known to be related to AD from the DMR results, showing the numbers of (**a**) hypermethylated genes and (**b**) hypomethylated genes in the BPA, BPF, and BPS exposure groups.

**Figure 2 ijms-25-01579-f002:**
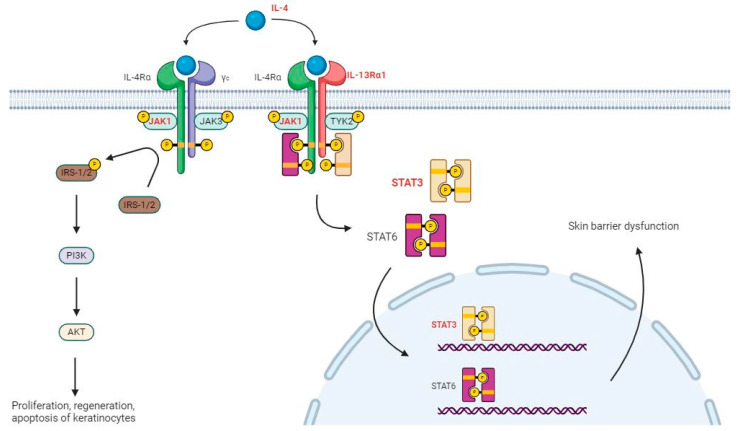
The JAK-STAT signaling pathway and PI3K-AKT signaling pathway that affect AD. Red text indicates the genes that are commonly included in the JAK-STAT and PI3K-AKT signaling pathways in all BPA, BPF, and BPS exposure groups from the DMR analysis. Hypomethylated genes can be hyperregulated, which can cause the proliferation, regeneration, apoptosis of keratinocytes in the PI3K-AKT signaling pathway, and skin barrier dysfunction in the JAK-STAT signaling pathway. Figure created using BioRender (https://biorender.com/, accessed on 12 October 2023).

**Table 1 ijms-25-01579-t001:** Information of participants.

Cohort Information	Atopy Group *n* = 19	Non-Atopy Group *n* = 98
**Age**	33.7 ± 3.5	33.9 ± 3.3
**Smoking**		
No	17 (89.5%)	91 (92.9%)
Yes	2 (10.5%)	7 (7.1%)
**BPA_exposure (μ** **g/g cr.)**	1.0 ± 0.8	1.7 ± 3.8
**BPF_exposure (μg/g cr.)**	0.4 ± 0.6	0.8 ± 4.0
**BPS_exposure (μ** **g/g cr.)**	0.3 ± 0.7	0.3 ± 0.7

**Table 2 ijms-25-01579-t002:** DMR results according to exposure group.

Personal Characteristic	Case Group	Control Group
**BPA (*n*)**	*n* = 3	*n* = 72
**BPA (μg/g cr.)**	2.5 ± 0.7	0.6 ± 0.4
**Age**	38.7 ± 3.2	33.9 ± 3.2
**BPF (*n*)**	*n* = 6	*n* = 75
**BPF (μg/g cr.)**	1.1 ± 0.8	0.1 ± 0.1
**Age**	32.5 ± 3.3	33.9 ± 3.4
**BPS (*n*)**	*n* = 5	*n* = 74
**BPS (μg/g cr.)**	1.2 ± 1.1	0.1 ± 0.0
**Age**	33.8 ± 5.1	33.5 ± 3.2

**Table 3 ijms-25-01579-t003:** Correlation analysis between variables (Spearman correlation).

Spearman	Age	Smoke	BPA Exposure	BPF Exposure	BPS Exposure
**Age**	1				
**Smoke**	0.226	1			
**BPA_exposure**	−0.016	−0.064	1		
**BPF_exposure**	0.036	−0.152	0.175	1	
**BPS_exposure**	0.104	0.158	−0.073	−0.032	1

**Table 4 ijms-25-01579-t004:** Correlation analysis between variables (*p*-value).

*p*-Value	Age	Smoke	BPA Exposure	BPF Exposure	BPS Exposure
**Age**	-				
**Smoke**	0.016	-			
**BPA_exposure**	0.869	0.495	-		
**BPF_exposure**	0.706	0.101	0.059	-	
**BPS_exposure**	0.273	0.089	0.435	0.734	-

**Table 5 ijms-25-01579-t005:** DMR results of hypermethylated regions and hypomethylated regions.

Material	Total DMRs	Hypermethylated	Hypomethylated
**BPA**	80,072	81	79,991
**BPF**	92,573	160	92,413
**BPS**	61,367	3056	58,311

**Table 6 ijms-25-01579-t006:** The number of genes related to atopy among the DMR results.

Material	Total Genes	Hypermethylated	Hypomethylated
**BPA**	390	1	389
**BPF**	356	2	354
**BPS**	303	14	289

**Table 7 ijms-25-01579-t007:** List of genes involved in the PI3-AKT and JAK-STAT signaling pathways from DMR analysis. The genes that are commonly included in the BPA, BPF, and BPS exposure groups are listed.

Material	PI3-AKT Signaling Pathway	JAK-STAT Signaling Pathway
**BPA, BPF, BPS**	MTOR, TP53, NTRK1, LAMC2, JAK1, JAK2, EGFR, NFKB1, YWHAE, KITLG, SYK, BDNF, IL4, BCL2, IL7, FN1, ANGPT2	MTOR, IL12RB1, EGF, JAK1, STAT3, IL13RA1, SOCS5, IL12RB2, EGFR, IL21R, IL4, IL15RA, BCL2, STAT5B, IL10RB, PIAS1, STAT1, IL7
**BPA, BPF**	NTF4	IL13RA2, IL33, TYK2, IL23R
**BPA, BPS**	IL4R, IL2RA, JAK3, IL6R	IL4R, IL2RA, JAK3, IL15, IL6ST, IL6R, STAT5A
**BPF, BPS**	IL2RB	IL23A, IL2RB
**BPA**	FLT4	IL21, IL24, IL31RA
**BPF**	ERBB2, CSF1, RELA	
**BPS**	NGFR	

## Data Availability

The data presented in this study are available on request from the corresponding author.

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
