# Peer review of "Epigenetic Methylation Changes in Pregnant Women: Bisphenol Exposure and Atopic Dermatitis"

_ijms, 2024, doi:10.3390/ijms25031579_

Round 1

Reviewer 1 Report (New Reviewer)

Comments and Suggestions for Authors

Please, find attached my review.

Comments on the Quality of English Language

Author Response

  1. I just want to ask the authors to clarify the term “atopic diseases” they have used in line 51.

A: Changed “atopic diseases” to “AD” to match the topic of this study to avoid confusion as suggested.

Reviewer 2 Report (New Reviewer)

Comments and Suggestions for Authors

The work is dedicated to the influence of different types of bisphenols (BP - BPA, BPF, BPS) on epigenetic DNA modification (methylation) in pregnant women with atopic dermatitis (AD) compared to the control group (without AD).

Bisphenols, chemicals that disrupt the endocrine system, are widely used in everyday life. Continued exposure to BPs during key periods of life development (pregnancy, infancy and early childhood) may contribute to adverse health effects such as reduced lung function, asthma, allergies and changes in immune system responses. In particular, data from several systematic reviews show an association between elevated urinary BPA glucoronide concentrations and allergic status, including blood levels of IgE. In addition, based on the results of the meta-analysis, Tang N et al. found a positive association between BPA exposure during pregnancy and the occurrence of allergic diseases in childhood. Therefore, the topic of the article is relevant.

The main drawbacks of the article:

1. There is currently no doubt that BPA can cause epigenetic changes in various cell types, including changes in the differentially methylated regions (DMRs) of various genes. At the same time, increased methylation in the promoter region of the gene can lead to inhibition of its transcription, and decreased methylation in the body region of the gene can be associated with increased gene expression. Thus, unidirectional changes in methylation in different regions of a gene can lead to multidirectional results regarding the expression of that gene. The authors clearly did not take these epigenetic patterns into account when discussing their own research.

2. The authors used the main group of women with AD (n-19) and the control group or pregnant women without AD (n-98). At the same time, comparable urinary concentrations of BPA, BPF and BPS were found in all groups during a single study. Furthermore, the authors randomly divided each group into 3 subgroups, namely those with relatively high levels of BPA (n-3), BPF (n-6), BPS (n-5) in the main group. Conversely, in the control group, the results with the lowest concentrations of BPA, BPF and BPS (n-72-75) were selected as subgroups. The authors found differences in DMRs between these AD/non-AD subgroups. In my opinion, the number of observations in the AD subgroups, especially BPA, is not representative. The study design used does not allow the effect of BP to be distinguished from the effect of AD pathogenetic factors. In addition, the effect of relatively low concentrations of BPA in the BPF and BPS subgroups cannot be excluded. Therefore, the study design does not meet the objectives of this study.

3. When interpreting their research results, the authors do not take into account that AD is a heterogeneous disease in its etiology and pathogenesis. Currently, AD is divided into two main subtypes [Tokura Y, Hayano S. Subtypes of atopic dermatitis: From phenotype to endotype. Allergol Int. 2022 Jan;71(1):14-24. doi: 10.1016/j.alit.2021.07.003. Epub 2021 Jul 31. PMID: 34344611]: (1) "External" - about 80% of the total amount of AD, is associated with a violation of the barrier function of the skin, the immune response of the second type (Th2, IgE) to protein allergens prevails and (2) "Internal" - women predominate, the barrier function of the skin is preserved, the immune response of the first dominates type (Th1) for metals or unknown antigens/haptens. In addition, Asian patients with AD are characterized by a unique phenotype of mixed immune dysregulation and barrier features between patients with AD and patients with psoriasis, with a high level of immune response of the third type (Th17/IL-17) [Tokura Y, Hayano S. Subtypes of atopic dermatitis: From phenotype to endotype. Allergol Int. 2022 Jan;71(1):14-24. doi: 10.1016/j.alit.2021.07.003. Epub 2021 Jul 31. PMID: 34344611]. These differences must also be taken into account when evaluating the epigenetics of AD.

Conclusion. Despite the relevance of the topic, the design and results of the study do not correspond to the goals and conclusions of this work. This article is incomplete and may mislead the readers of the journal.

Author Response

  1. There is currently no doubt that BPA can cause epigenetic changes in various cell types, including changes in the differentially methylated regions (DMRs) of various genes. At the same time, increased methylation in the promoter region of the gene can lead to inhibition of its transcription, and decreased methylation in the body region of the gene can be associated with increased gene expression. Thus, unidirectional changes in methylation in different regions of a gene can lead to multidirectional results regarding the expression of that gene. The authors clearly did not take these epigenetic patterns into account when discussing their own research.

A: Changes in gene expression levels occur after epigenetic changes, so we focused on epigenetic changes and verified whether methylation-changed genes belong to pathways associated with atopic dermatitis using KEGG pathway analysis. Hypomethylated genes are related to JAK-STAT and PI3K-AKT signaling pathways, which are important for the pathophysiology of atopic dermatitis. We showed this in Figure 2 and Table 7 of the paper. This analysis indicates that the methylation changes found in this study considered both gene expression and the relevance of atopic dermatitis. Also, we are aware of the importance of genomic patterns suggested by the reviewer, and we plan to conduct a multiomics analysis as a follow-up study.

  1. The authors used the main group of women with AD (n-19) and the control group or pregnant women without AD (n-98). At the same time, comparable urinary concentrations of BPA, BPF and BPS were found in all groups during a single study. Furthermore, the authors randomly divided each group into 3 subgroups, namely those with relatively high levels of BPA (n-3), BPF (n-6), BPS (n-5) in the main group. Conversely, in the control group, the results with the lowest concentrations of BPA, BPF and BPS (n-72-75) were selected as subgroups. The authors found differences in DMRs between these AD/non-AD subgroups. In my opinion, the number of observations in the AD subgroups, especially BPA, is not representative. The study design used does not allow the effect of BP to be distinguished from the effect of AD pathogenetic factors. In addition, the effect of relatively low concentrations of BPA in the BPF and BPS subgroups cannot be excluded. Therefore, the study design does not meet the objectives of this study.

A: Thank you for your comment. We think that the low-exposure group can represent the low exposure level because the LOD values are included in the bisphenol detection values of the cohort. And We think that the interaction between each substance exposure can be excluded because the independence of BPA, BPF, and BPS exposure levels was verified using SPSS (Ver. 27). In response to your suggestions, we have revised the Materials and Methods section as follows (page 8, lines 266-279):

In this study, we used the exposure levels of BPA, BPF, and BPS measured in the cohort. The exposure levels of BPA, BPF, and BPS may be related to the occurrence of AD; therefore, we classified them into control and case groups according to the exposure levels of each substance. The upper 25% of the exposure level was classified as a high-exposure group and the lower group as a low-exposure group (BPA: 1.685 ug/g creatinine, BPF: 0.445 ug/g creatinine, and BPS: 0.178 ug/g creatinine). For each substance, individuals with AD and belonging to the high-exposure group were designated as case groups and individuals without AD and belonging to the low-exposure group were designated as control groups. In this classification, the low-exposure group also included values that the device could not measure; >Limit of Detection (LOD), and >LOD was calculated using Equation 1 according to the reference of Korea Occupational Safety and Health Agency (KOSHA). The LOD values of BPA, BPF, and BPS were 0.052, 0.008, and 0.025, respectively.

(1)

According to the equation, the >LOD values of BPA, BPF, and BPS were replaced with 0.03677 μg/g cr., 0.056569 μg/g cr., and 0.017678 μg/g cr., respectively.

  1. When interpreting their research results, the authors do not take into account that AD is a heterogeneous disease in its etiology and pathogenesis. Currently, AD is divided into two main subtypes [Tokura Y, Hayano S. Subtypes of atopic dermatitis: From phenotype to endotype. Allergol Int. 2022 Jan;71(1):14-24. doi: 10.1016/j.alit.2021.07.003. Epub 2021 Jul 31. PMID: 34344611]: (1) "External" - about 80% of the total amount of AD, is associated with a violation of the barrier function of the skin, the immune response of the second type (Th2, IgE) to protein allergens prevails and (2) "Internal" - women predominate, the barrier function of the skin is preserved, the immune response of the first dominates type (Th1) for metals or unknown antigens/haptens. In addition, Asian patients with AD are characterized by a unique phenotype of mixed immune dysregulation and barrier features between patients with AD and patients with psoriasis, with a high level of immune response of the third type (Th17/IL-17) [Tokura Y, Hayano S. Subtypes of atopic dermatitis: From phenotype to endotype. Allergol Int. 2022 Jan;71(1):14-24. doi: 10.1016/j.alit.2021.07.003. Epub 2021 Jul 31. PMID: 34344611]. These differences must also be taken into account when evaluating the epigenetics of AD.

A: Thank you for your opinion on the importance of considering AD subtypes in the heterogeneity and epigenetic analysis of AD. However, all subtypes share a similar inflammatory pattern associated with Th2 cytokines, and intrinsic AD and extrinsic AD are not mutually exclusive, and intrinsic AD can develop into extrinsic AD over time. Also, since subtypes were not distinguished epigenetically, we conducted an analysis on comprehensive atopic dermatitis in this study. We also added the following content to the Introduction section based on your suggestion (page 2, lines 69-75):

AD is a heterogeneous disease that shows varied clinical phenotypes, and can be classified into extrinsic and intrinsic subtypes depending on the morphology and distribution of lesions, serum IgE level, type of allergic reaction, etc. Distinguishing AD subtypes can help understand and improve the etiology, diagnosis, prognosis, treatment, and management of AD. However, AD is generally known as a Th2-based disease, and the distinction between subtypes is not permanent as intrinsic AD can become extrinsic AD.

This manuscript is a resubmission of an earlier submission. The following is a list of the peer review reports and author responses from that submission.

Round 1

Reviewer 1 Report

Comments and Suggestions for Authors

The authors aimed to examine the epigenetic changes related to bisphenols (BPA, BPF, BPS) exposure and atopic dermatitis (AD) in PBMC of pregnant women.

However, the study design and analytical strategies employed do not allow to discriminate the effect of any bisphenol from AD as these variables are completely confounded by each other in the selection of 'case' and 'control' groups. Indeed, the 2 groups compared are always: 'high bisphenol in AD women' versus 'low bisphenol in non-AD women'. Any change resulting from this comparison is likely to be driven by the impact of an already established AD and not bisphenol, or by any other confounders that the authors did not take into consideration, the main ones being cell type composition, age and smoking status which might greatly vary between groups. The fact that the results from this comparison overlap with AD genes is therefore not surprising at all. Additionally, the effects of different bisphenols cannot be separated from each other either unless high BPA mean low BPF and BPS (and vice versa), which has not been investigated.

Such a flawed design/analysis drastically impairs the proper interpretation of the data and induces an obvious bias in the conclusions drawn by the authors. Instead, an appropriate group design, comparison and analytical modeling is required to derive any 'clean' findings.

Reviewer 2 Report

Comments and Suggestions for Authors

Kim et al. focused on relationship among bisphenol exposure, Atopic Dermatitis (AD), and DNA methylation. They found that genes belonging to the JAK-STAT and PI3K-AKT signaling pathways were hypomethylated in the blood of pregnant women exposed to bisphenols. However, their results are descriptive and difficult to interpret. In addition, important technical information and raw data are missing. Therefore, to review the manuscript adequately, the authors must show more information and data.

1) Table 2: The number of case groups are limited. So, it is difficult to make conclusion.

2) Table 3: The summarized information (results) of DMR regions is necessary as supporting information. This information is quite important to interpret results.

3) Typical examples of hypomethylated regions should be shown as figures.

4) Figure 2 is confusing. JAK-STAT and PI3-AKT are important signaling for a lot of cell types. Even if some related genes are hypomethylated, it does not make sense to directly connect them with AD. No convincing results.

5) Almost all references are review articles but not original research papers.